# Structural basis of superinfection exclusion by bacteriophage T4 Spackle

Ke Shi[1,2], Justin T. Oakland[1,2], Fredy Kurniawan[1,2], Nicholas H. Moeller[1,2], Surajit Banerjee[3] & Hideki Aihara [1,2 ✉]

A bacterial cell infected with T4 phage rapidly establishes resistance against further infections by the same or closely related T-even-type bacteriophages – a phenomenon called superinfection exclusion. Here we show that one of the T4 early gene products and a periplasmic protein, Spackle, forms a stoichiometric complex with the lysozyme domain of T4 tail spike protein gp5 and potently inhibits its activity. Crystal structure of the Spackle-gp5 lysozyme complex shows that Spackle binds to a horseshoe-shaped basic patch surrounding the oligosaccharide-binding cleft and induces an allosteric conformational change of the active site. In contrast, Spackle does not appreciably inhibit the lysozyme activity of cytoplasmic T4 endolysin responsible for cell lysis to release progeny phage particles at the final step of the lytic cycle. Our work reveals a unique mode of inhibition for lysozymes, a widespread class of enzymes in biology, and provides a mechanistic understanding of the T4 bacteriophage superinfection exclusion.

[1] Department of Biochemistry, Molecular Biology and Biophysics, University of Minnesota, 321 Church Street S.E., Minneapolis, MN 55455, USA. [2] Institute for Molecular Virology, University of Minnesota, Minneapolis, MN 55455, USA. [3] Northeastern Collaborative Access Team, Cornell University, Advanced Photon Source, Lemont, IL 60439, USA. ✉email: aihar001@umn.edu

The lifecycle of bacteriophage T4 starts with the attachment of phage virion to host *Escherichia coli* surface, which entails reversible binding of the phage's long tail fibers to bacterial outer membrane protein C (OmpC) or lipopolysaccharide, followed by irreversible adsorption of the short tail fibers to the bacterial surface[1,2]. The attachment triggers a conformational change of the baseplate at the distal end of the phage tail tube and contraction of tail sheath, revealing a puncturing device for penetration of the bacterial cell envelope and delivery of the viral DNA[3,4]. The T4 spike protein gp5 (gene product 5) located at the tip of this puncturing device has an internal lysozyme domain and the C-terminal domain that is folded into a trimeric β-helix and functions as a needle to penetrate the bacterial outer membrane[3,5,6] (Supplementary Fig. S1). A subsequent release of the β-helix domain from the baseplate activates gp5 lysozyme activity, which degrades peptidoglycan in the bacterial cell wall[7]. Lastly, the tail tube fuses with a remodeled bacterial inner membrane via a less well-understood mechanism to promote the release of the viral DNA genome into the bacterial cytoplasm[8].

One of the T4 early gene products produced immediately following the genome injection is gp61.3, also known as Spackle. Spackle, along with another T4 gene product Imm, confers the infected host *E. coli* with resistance against secondary phage infections—a phenomenon known as "superinfection exclusion"[9–12]. Genetic studies have led to a hypothesis that Spackle achieves superinfection exclusion by inhibiting the gp5 lysozyme activity in the periplasm of infected bacteria and thereby preventing the penetration of the tail tube of incoming phages[13–18]. Our recent studies have shown that the mature, 75-amino acid Spackle protein after proteolytic cleavage of the periplasmic localization signal has a compact α-helical bundle fold with a disulfide bond, consistent with its presumed function in the periplasm[19]. However, it is currently unknown whether Spackle actually interacts with the tail (gp5)

lysozyme or inhibits its glycoside hydrolase activity. Here, we address this question through structural and biochemical studies. Our results show that T4 Spackle forms a stoichiometric complex with the lysozyme domain of gp5 and potently inhibits its enzymatic activity, whereas Spackle does not appreciably inhibit the cytoplasmic T4 lysozyme (endolysin, gpe) responsible for cell lysis at the final step of the phage lytic cycle. Crystal structure of the Spackle–gp5 complex shows that the binding of Spackle induces an allosteric conformational change in the gp5 lysozyme active site and partially blocks its oligosaccharide-binding cleft. Our work reveals a mode of lysozyme inhibition distinct from those used by the bacterial proteinaceous lysozyme inhibitors and provides mechanistic understanding of the T4 bacteriophage superinfection exclusion.

## Results

**T4 Spackle binds gp5 lysozyme.** To test whether the tail lysozyme and Spackle directly interact, we expressed the lysozyme domain of T4 gp5, which spans residues Met174 to Glu342 of full-length gp5. Full-length mature Spackle (Gly23 to Glu97) was also purified as a soluble protein, as reported previously[19]. When the purified Spackle (9.7 kDa) and the lysozyme domain of gp5 (20.1 kDa, hereinafter referred to as gp5 lysozyme) were individually injected into a Superdex 75 size-exclusion chromatography (SEC) column, either protein alone eluted as single peaks with the apparent masses of 8 and 17 kDa based on elution volume, respectively, which are slightly smaller than their actual monomer molecular masses. On the other hand, when gp5 lysozyme and Spackle were mixed and co-injected into the SEC column, the two proteins co-eluted earlier than either protein alone, suggesting that they formed a stable complex (Fig. 1a). The apparent mass of the complex based on the elution volume of the SEC peak (~23 kDa) and relative band intensities in sodium dodecyl sulfate polyacrylamide gel electrophoresis (SDS-PAGE) for the two proteins co-eluted from the column was consistent with the

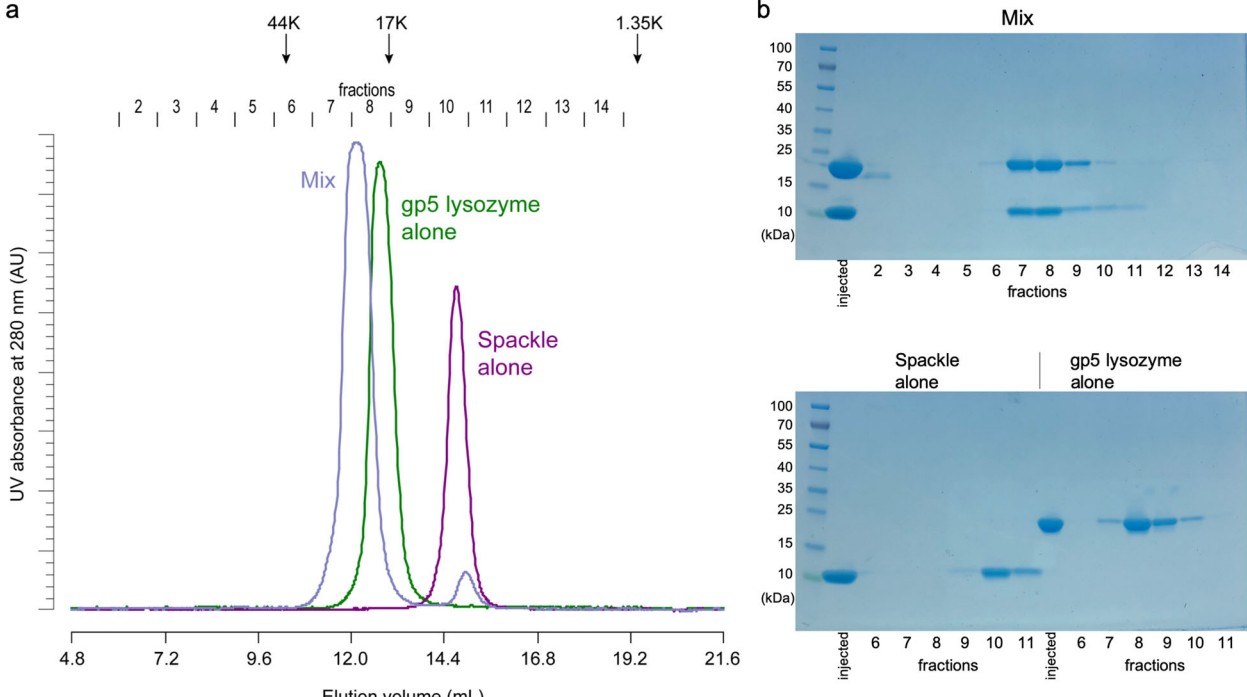

**Fig. 1 Spackle forms a stable complex with gp5 lysozyme. a** Overlaid SEC traces of Spackle alone, gp5 lysozyme alone, and the mixture of the two proteins. The elution positions of the size standards: ovalbumin (44 kDa), myoglobin (17 kDa), and vitamin B12 (1.35 kDa) are indicated at the top. The collection scheme for the fractionated proteins is shown above the chromatograms. **b** Coomassie blue-stained SDS-PAGE analysis of the fractionated proteins from the SEC runs in (**a**).

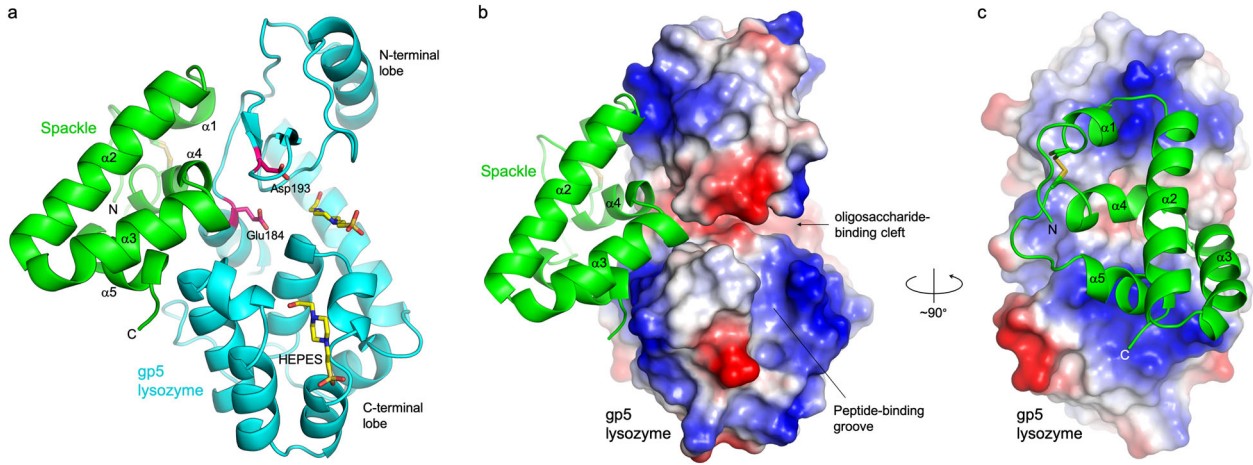

**Fig. 2 Overall structure of the gp5 lysozyme–Spackle complex. a** Ribbon model depicting the crystal structure of the complex. The catalytic residues of gp5 lysozyme (Glu184 and Asp193) and the HEPES molecules bound on the protein surface are shown as sticks. **b** Similar to **a**, with the molecular surface of gp5 lysozyme colored according to electrostatic potential (blue: positive, red: negative). **c** A view after ~90° rotation around a vertical axis from that in (**b**). Spackle binds to a basic horseshoe-shaped patch on the gp5 lysozyme surface, including Arg181, Arg182, Arg187, and Lys189 from the N-terminal lobe and Arg312, Lys321, and Arg326 from the C-terminal lobe.

complex containing a 1:1 molar ratio of gp5 lysozyme and Spackle (Fig. 1b).

**Crystal structure of the Spackle–gp5 lysozyme complex.** Given that Spackle and gp5 lysozyme form a stable complex in solution, we sought to understand the mechanism of this protein–protein interaction through structural analysis. To that end, we co-crystallized Spackle and gp5 lysozyme and determined the crystal structure of the complex by molecular replacement phasing. Two crystal forms were obtained, in monoclinic and orthorhombic lattice systems, for which the structures were refined to 1.78 and 1.92 Å resolution, respectively. As expected, the crystal structures show a 1:1 Spackle–gp5 lysozyme complex (Fig. 2a–c). The monoclinic crystal contained two Spackle–gp5 lysozyme complexes in the asymmetric unit, whereas the orthorhombic crystal contained one complex in the asymmetric unit. In addition, we observed several 4-(2-hydroxyethyl)-1-piperazineethanesulfonic acid (HEPES) molecules bound on the lysozyme surface, including one occupying the oligosaccharide-binding cleft (C-site)[20] and another in the peptide-binding groove in the monoclinic crystal structure (Fig. 2a). The protein structures of the three crystallographically independent complexes were essentially identical to each other, with pairwise backbone room mean square deviations within 0.5 Å.

**Spackle–gp5 lysozyme interaction.** Spackle binds to gp5 lysozyme primarily via three short helices (α1, α4, and α5) located on one side of its flat helical bundle fold. The interface buries a total of 952 Å² surface area and involves numerous polar contacts (Fig. 2a, c). Spackle engages a highly positively charged horseshoe-shaped surface around the deep substrate-binding cleft, spanning both the N and C-terminal lobes of lysozyme (Fig. 2b, c). The side chains of a number of basic amino acid residues of gp5 lysozyme, namely Arg181, Arg182, Arg187, and Lys189 from the N-terminal lobe and Arg312, Lys321, and Arg326 from the C-terminal lobe, are involved in intermolecular hydrogen bonds or salt bridges with Spackle (Fig. 3). Many main-chain atoms from either protein also make direct contacts at the interface. For instance, the main chains of Trp192 and Leu186 from gp5 lysozyme make bidentate hydrogen bonds with the side chain carboxamide moieties of Gln37 and Asn76 from Spackle, respectively; the main-chain carbonyl groups of Ser32, Met33,

Thr34, and Ala35 from Spackle at the C-terminus of α1 helix accept a cluster of hydrogen bonds from gp5 lysozyme (Fig. 3). Although Spackle binds over and partially blocks a distal site (E/F-site)[21] in the substrate-binding cleft of lysozyme, it does not block the enzyme active site or interact directly with the catalytic residues (Glu184 and Asp193) (Fig. 2a).

**Protein structural changes upon binding.** Superposition of Spackle and gp5 lysozyme structures in the complex with their respective unbound state structures shows conformational changes for both proteins upon binding. In the Spackle-bound gp5 lysozyme, compared to its structure in the full-length trimeric (gp27–gp5)₃ complex[22], both N and C-terminal lobes are rotated away from each other so as to widen the substrate-binding cleft (Fig. 4a). The conformational change displaces the lysozyme catalytic residues (Glu184 and Asp193), both of which are in the N-terminal lobe, with respect to the oligosaccharide-binding sites. The α4 and α5 helices of Spackle are stuck at the mouth of this widened cleft, which could stabilize its forced-open conformation (Fig. 2a, b). In the lysozyme-bound Spackle, compared to its free state[19], the short α1 helix is kinked in the middle toward Arg187, Leu188, and Lys189 of gp5 lysozyme, which donate hydrogen bonds to the residues at the C-terminal end of this helix (Fig. 3). This causes the residues N-terminal to the kink, Lys26 to Cys29, to lose their regular α-helical structure (Fig. 4b). Notably, the electron density for Cys29 and Cys81 of Spackle, which were observed to be disulfide-bonded in the unliganded structure, is best accounted for when the model was refined as a mixture of ~60% oxidized (disulfide-bonded) and ~40% reduced (Sγ–Sγ distance of 3.0 Å) species. The partial loss of disulfide linkage may reflect the conformational strain in α1 helix upon lysozyme binding and resulting bond breakage or preferential binding of Spackle in the reduced form during crystallization (Supplementary Fig. S2). These observations suggest that Spackle–gp5 lysozyme interaction is mediated by mutual conformational changes, likely via an induced-fit mechanism.

**Spackle inhibits gp5 lysozyme activity.** Based on the results described above, we next tested whether Spackle inhibits lysozyme enzyme activities, using a fluorescence-based assay to monitor the degradation of fluorescein-labeled *Micrococcus lysodeikticus* cell walls over time. In a titration experiment, adding

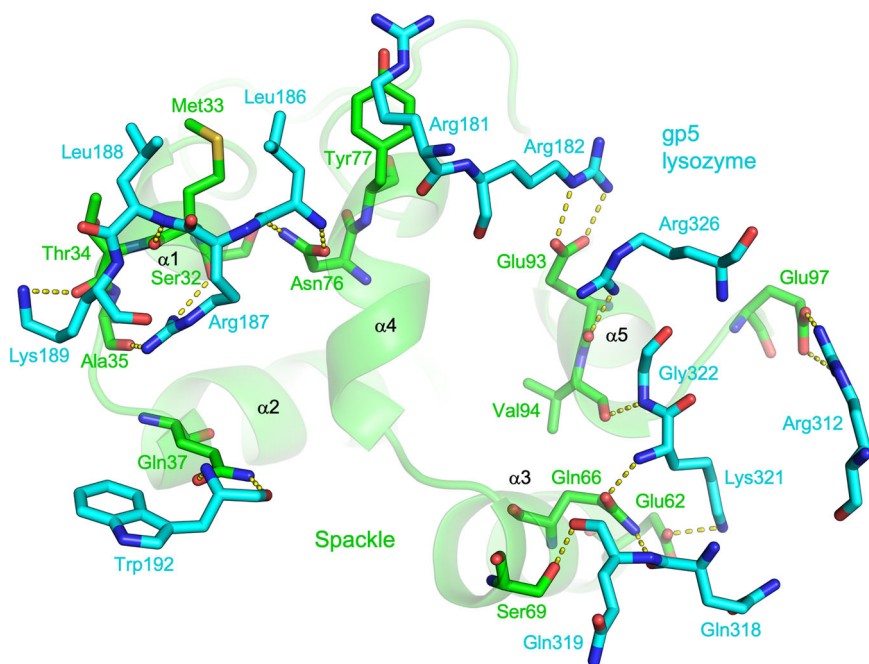

**Fig. 3 Spackle–gp5 lysozyme interface.** Amino acid residues of Spackle (green) and gp5 lysozyme (cyan) involved in direct interaction are shown as sticks, with a semi-transparent backbone ribbon model of Spackle in the background. The intermolecular hydrogen bonds and salt bridges are indicated by yellow dashed lines.

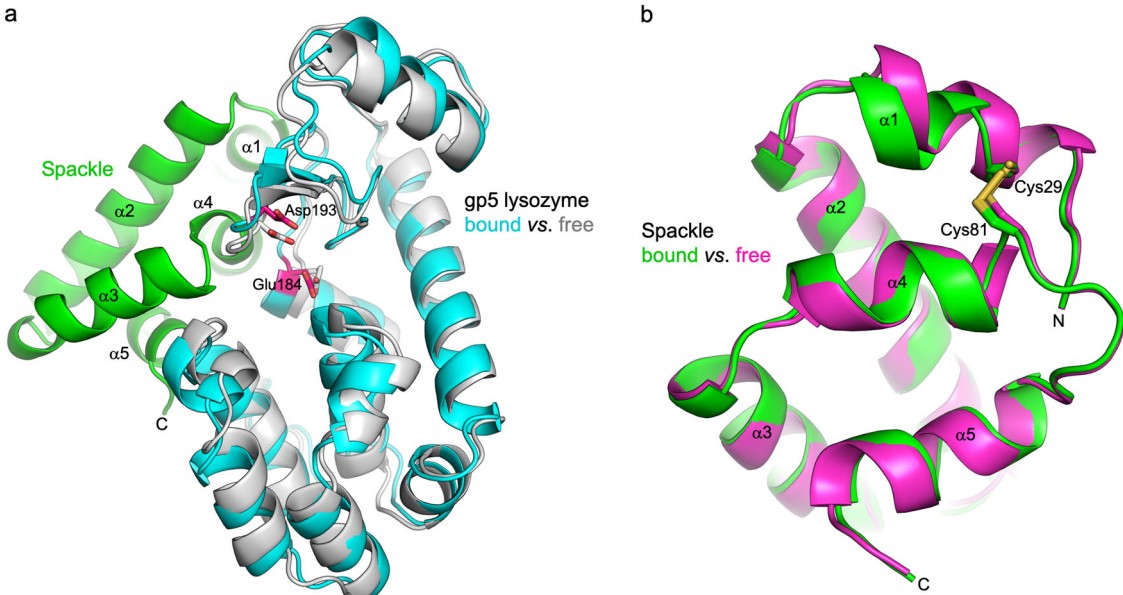

**Fig. 4 Conformational changes upon binding. a** Superposition between free (gray, PDB ID: 1WTH) and Spackle-bound (cyan) gp5 lysozyme structures, with the catalytic residues shown as sticks. **b** Superposition between free (magenta, PDB ID: 6X6O)[19] and gp5-bound (green) Spackle structures, with the cysteine side chains shown as sticks.

Spackle at various molar ratios to gp5 lysozyme at a fixed concentration caused a dose-dependent inhibition of the lysozyme activity (Fig. 5a), with weak residual activity observed at equimolar ratio and completely no activity at twofold excess of Spackle. On the other hand, in parallel experiments, Spackle also affected the activity of cytoplasmic T4 lysozyme (hereinafter referred to as endolysin), which shares a 43% sequence identity with the lysozyme domain of gp5[23] (Figs. 5b and 6). However, the addition of Spackle did not alter the rate of product accumulation by endolysin and instead led to a longer initial lag phase, the reason for which is unknown. In any case, the inhibition of endolysin by Spackle was weak, if any, where we still observed enzyme activity even at fourfold excess of Spackle over endolysin (Fig. 5b). Consistently, we did not detect a stable complex formation between endolysin and Spackle by SEC (Supplementary Fig. S3) as we did for gp5 lysozyme and Spackle (Fig. 1). Lastly, we observed no inhibitory effect of Spackle on hen egg-white lysozyme, which shares the same enzymatic activity but no

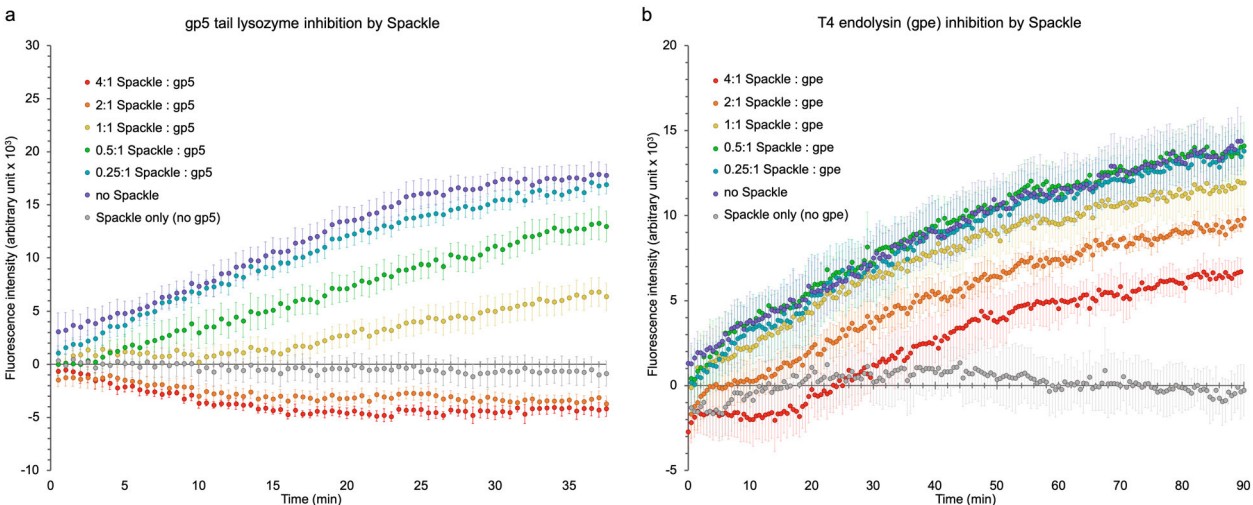

**Fig. 5 Selective inhibition of gp5 lysozyme activity by Spackle. a** Cell wall degrading activity of gp5 lysozyme measured in the presence of varying amounts of Spackle. **b** Cell wall degrading activity of the cytoplasmic T4 lysozyme, endolysin, measured in the presence of varying amounts of Spackle. In both graphs the average of quadruplicate experiments ± standard deviation is plotted.

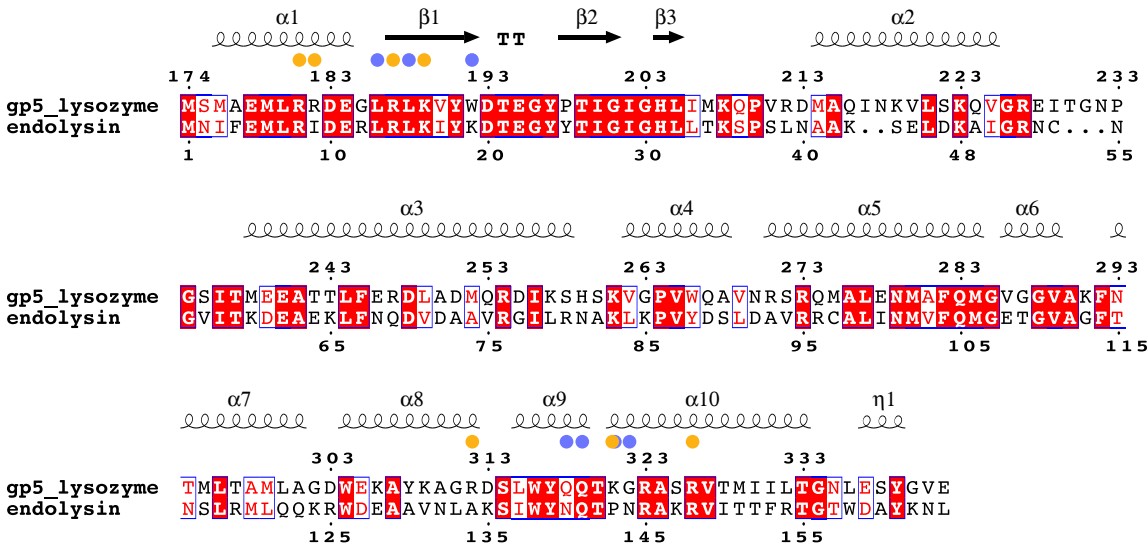

gp5 residues whose side chain (🟡) or main chain (🔵) is involved in hydrogen-bond or salt bridge interaction with Spackle are marked.

**Fig. 6 Amino acid sequence alignment between T4 gp5 lysozyme and endolysin.** This figure was generated using ESPript[38].

appreciable sequence homology with either T4 lysozyme (Supplementary Fig. S4). Collectively, these results show that Spackle is a selective inhibitor of gp5 lysozyme.

## Discussion

We found in this study that T4 Spackle forms a stoichiometric complex with the tail spike (gp5) lysozyme and inhibits its enzymatic activity. T4 *spackle* gene was first identified for the phenotype of its mutation to permit the lysis of bacteria in the absence of T4 endolysin due to a defect in the resistance to "lysis from without"[13], which is a mechanism of cell lysis mediated by the adsorption of many phage particles[24]. Mutations in *spackle* cause truncation of the latency period known as "lysis-inhibition collapse"[25] and a partial loss of superinfection exclusion[9,11], consistent with the role of Spackle in inhibiting cell lysis. The original S12 mutation in *spackle* leads to the replacement of nine amino acid residues from the C-terminus of Spackle with an

unrelated sequence[13,15,18,25]. These residues correspond to α5 helix and the following C-terminal tail, which are shown here to make extensive interaction with gp5 lysozyme (Figs. 2 and 3). Notably, a temperature-sensitive pseudo-revertant of an amber mutation in gene *e* was isolated and the reversion mutation was mapped in gene 5 (5ts1), which leads to a Gly322 to Asp amino acid substitution in gp5 lysozyme[16,26]. Our structural data provide a ready explanation (Supplementary Fig. S5)—the G322D substitution would compromise the interaction of gp5 with Spackle by causing a steric clash and therefore let its lysozyme activity complement the loss of gpe/endolysin. Thus, our results rationalize the earlier genetic studies and further support the hypothesis that Spackle functions in the periplasm of T4 phage-infected *E. coli* cells to inhibit the activity of phage tail lysozyme. An alignment of a total of 178 Spackle homologs from various bacteriophages available in the UniProt database shows generally high sequence conservation, and especially for the residues

involved in direct contact with gp5 lysozyme, suggesting a conserved mode of function for these proteins (Supplementary Fig. S6).

Interestingly, while Spackle potently inhibits gp5 lysozyme, it binds to a region distal to the active site and does not directly engage the catalytic residues. Our crystal structures suggest that the inhibition by Spackle is mediated by an allosteric rearrangement of the lysozyme active site and a partial blockage of its oligosaccharide-binding cleft. It is notable that the same side of the oligosaccharide-binding cleft is blocked by the β-helix domain in the full-length trimeric gp5–gp27 complex[22] (Supplementary Fig. S1). This mechanism is in contrast to the mode of lysozyme inhibition by other known proteinaceous lysozyme inhibitors, including bacterial Ivy, MliC, PliC, PliG, and PliI, which consist primarily of β-sheet and insert a loop between antiparallel β-strands into the active site to directly engage the lysozyme catalytic residues[27–32] (Supplementary Fig. S7). The distinct α-helical bundle structure and unique inhibition mechanism of Spackle suggest that it is evolutionarily independent of the bacterial lysozyme inhibitor proteins.

We observed that the cytoplasmic T4 lysozyme, endolysin, is not inhibited by Spackle, despite its high similarity to gp5 lysozyme (Fig. 5). As mentioned above, T4 gp5 lysozyme and endolysin are highly homologous to each other and share a 43% amino acid sequence identity. Thus, it is not surprising that many of the residues involved in the interaction with Spackle are conserved. However, several key basic residues of gp5 are not conserved; Arg182, Arg312, and Lys321 of gp5 align with non-polar residues Ile9, Ala134, and Pro143 of endolysin (Fig. 6). In addition, the main-chain amide group of Lys321 and Gly322 of gp5 are involved in hydrogen-bonding with Spackle (Fig. 3), where the corresponding endolysin residues Pro143 and Asn144 are not compatible with the same interaction due to the lack of amide hydrogen and clash of the Asn side chain (analogous to gp5 G322D mentioned above), respectively. These differences are likely to hamper optimal interaction between Spackle and T4 endolysin, consistent with the lack of stable complex formation in our binding experiment by SEC (Supplementary Fig. S3) and leading to the weaker if any, inhibition. Given also that endolysin is one of the T4 late gene products that function at the last stage of the phage lytic cycle[33], it seems unlikely that Spackle, one of the early gene products, would physiologically interact with endolysin.

## Methods

**Protein purification.** T4 Spackle, gp5 tail lysozyme, and T4 endolysin (gp*e* lysozyme) were recombinantly expressed in *E. coli* strain BL21(DE3) with a C-terminal 6×His-tag from pET-24a-based expression vectors and purified using nickel-affinity and SEC. In all cases the lysis of *E. coli* was carried out only by sonication and no lysozyme was added. The purified proteins in storage buffer containing 20 mM Tris-HCl pH 7.4, 0.5 M NaCl, and 5 mM β-mercaptoethanol (only for T4 endolysin, as Spackle and gp5 lysozyme have no free Cys), were flash-frozen in liquid nitrogen and stored at −80 °C.

**Protein crystallization and structure determination.** Purified Spackle and gp5 lysozyme were mixed in a 1:1 molar ratio to yield a complex of ~0.6 mM (~18 mg ml⁻¹). The Spackle–gp5 lysozyme crystals were grown by the hanging-drop vapor diffusion method by mixing the complex solution with an equal volume of a reservoir solution consisting of 35% (w/v) polyethylene glycol 3350, 20% 2-propanol, 0.1 M HEPES–NaOH pH 7.5. The crystals, which were long but thin rod-shaped, were directly flash-cooled by plunging into liquid nitrogen. X-ray diffraction data were collected at the Northeastern Collaborative Access Team (NE-CAT) beamlines of the Advanced Photon Source (APS, Lemont, IL) and processed using XDS[34]. The structure of the Spackle–gp5 lysozyme complex was determined by molecular replacement phasing by PHASER[35], using the crystal structures of free Spackle (PDB ID: 6X6O) and gp5 lysozyme (PDB ID: 1WTH)[22] as the search models. Iterative model building and refinement were performed using *Coot*[36] and PHENIX[37], respectively. A summary of data collection and model refinement statistics is shown in Table 1. Ramachandran plot shows favorable backbone geometries with 98.5% and 1.5% in the favored and allowed regions, respectively,

**Table 1 Data collection and refinement statistics.**

| PDB ID | 6XC0 | 6XC1 |
|---|---|---|
| *Data collection* | | |
| Space group | P2₁ | P2₁2₁2₁ |
| *Cell dimensions* | | |
| a, b, c (Å) | 41.31, 127.60, 46.09 | 41.08, 46.17, 129.50 |
| α, β, γ (°) | 90, 92.25, 90 | 90, 90, 90 |
| Resolution (Å) | 63.8–1.78 (1.85–1.78) | 39.2–1.92 (1.99–1.92) |
| $R_{merge}$ | 0.1145 (0.8976) | 0.09337 (1.609) |
| I/σI | 13.44 (1.73) | 11.21 (0.77) |
| Completeness (%) | 98.35 (96.99) | 98.80 (98.74) |
| Redundancy | 5.0 (5.0) | 5.1 (4.9) |
| $CC_{1/2}$ | 0.995 (0.633) | 0.997 (0.358) |
| *Refinement* | | |
| Resolution (Å) | 63.8–1.78 (1.85–1.78) | 39.2–1.92 (1.99–1.92) |
| No. of reflections | 44,705 (4388) | 19,321 (1888) |
| $R_{work}/R_{free}$ | 0.1560/0.1983 | 0.1861/0.2148 |
| No. of atoms | 4486 | 2098 |
| Protein | 3914 | 1959 |
| Ligand/ion | 75 | 20 |
| Water | 497 | 119 |
| B-factor | 29.12 | 45.62 |
| Protein | 27.34 | 45.09 |
| Ligand/ion | 55.07 | 65.43 |
| Water | 39.24 | 50.97 |
| R.m.s. deviations | | |
| Bond lengths (Å) | 0.013 | 0.003 |
| Bond angles (°) | 1.24 | 0.56 |

Statistics for the highest-resolution shell are shown in parentheses. Each structure is based on a single-crystal.

for the monoclinic crystal form (6XC0), and 97.9 and 2.1% for the orthorhombic (6XC1) crystal form.

**Protein–protein binding studies.** A 500 µL sample containing approximately 0.2 mM of either Spackle or gp5 lysozyme, or an equimolar mixture of both proteins, was injected into Superdex 75 10/300 SEC column. The column was operated at 4 °C with a flow rate of 0.4 ml min⁻¹, and the elution buffer contained 20 mM Tris-HCl pH 7.4 and 0.5 M NaCl. Protein complex formation was assessed by monitoring the elution profiles and analyzing the collected fractions by SDS-PAGE. The SEC column was calibrated using gel filtration standard (Bio-Rad) including thyroglobulin (670 kDa), γ-globulin (158 kDa), ovalbumin (44 kDa), myoglobin (17 kDa), and vitamin B12 (1.35 kDa). The interaction between T4 cytoplasmic (e) lysozyme and Spackle was tested in the same fashion. Uncropped gel images for data shown in Fig. 1 and Supplementary Fig. S3 are provided in Supplementary Fig. S8.

**Lysozyme activity assay.** The EnzCheck® Lysozyme Assay kit (Molecular Probes) was utilized to measure lysozyme activity in solution. The substrate was comprised of *Micrococcus lysodeikticus* cell walls, labeled with fluorescein to a degree that fluorescence is quenched. Lysozymes cleave β-(1,4) glycosidic bond between N-acetylmuramic acid and N-acetylglucosamine moieties in peptidoglycan and relieve the quenching, yielding fluorescence proportional to lysozyme activity. A 50 µg mL⁻¹ working suspension of the lysozyme substrate was prepared in a reaction buffer (0.1 M sodium phosphate pH 7.5, 0.1 M NaCl, 2 mM sodium azide). Reactions were performed in 96 chimney-well, flat bottomed black microplates with a final volume of 100 µL. Spackle protein was serially diluted across six wells with storage buffer achieving final concentrations of 104 µM, 52, 26, 13, 6.5, and 0 µM. Each lysozyme was added to 26 µM. To compensate for lower protein concentrations in some wells and the controls, 15 µM (1 mg mL⁻¹, final concentration) of bovine serum albumin (BSA) was added to all wells. All proteins were mixed first in 50 µL storage buffer and incubated at room temperature for 15 min. A volumetric equivalent of 50 µg mL⁻¹ lysozyme substrate, prepared as described above, was then added to each well before manually transferring the plate to the plate reader and immediately beginning fluorescence scan. Control samples contained Spackle (104 µM), BSA, and the substrate, but no lysozymes. Another set of control samples containing only BSA and the substrate was utilized to establish baseline fluorescence. Fluorescence intensity was measured using a Tecan Spark 10M microplate reader under a standard fluorescein setting: excitation/emission wavelengths of 494/521 nm. Wells were incubated at 35 °C and scanned every 30 s to monitor the kinetics of lysozyme

activity. All samples including the controls were quadruplicated. Following scans, the average fluorescence of the BSA and substrate only samples was determined to be the baseline fluorescence and was subtracted from all experimental samples. The results were plotted as the average ± standard deviation.

**Statistics and reproducibility.** The source data underlying Fig. 5 and Supplementary Fig. S4 are provided as supplementary data in the Excel spreadsheets. All samples were quadruplicated and the graphs show the average ± standard deviation for each data point.

**Reporting summary.** Further information on research design is available in the Nature Research Reporting Summary linked to this article.

## Data availability

The coordinates and structure factors for the crystal structures reported in this paper have been deposited in the Protein Data Bank with accession codes 6XC0 and 6XC1.

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

## Acknowledgements

This work was supported by a grant from the US National Institutes of Health (NIGMS R35-GM118047 to H.A.). This work is based upon research conducted at the Northeastern Collaborative Access Team beamlines, which are funded by the US National Institutes of Health (NIGMS P30 GM124165). The Pilatus 6M detector on 24-ID-C beamline is funded by a NIH-ORIP HEI grant (S10 RR029205). This research used resources of the Advanced Photon Source, a U.S. Department of Energy (DOE) Office of Science User Facility operated for the DOE Office of Science by Argonne National Laboratory under Contract No. DE-AC02-06CH11357, and those of the Minnesota Supercomputing Institute.

## Author contributions

K.S. collected X-ray diffraction data and determined the crystal structures. J.T.O. conducted lysozyme activity assays. K.S., J.T.O., F.K., and N.H.M. purified, analyzed the interactions of, and crystallized the proteins. S.B. collected X-ray diffraction data. H.A. managed the project and wrote the paper with input from all authors.

## Competing interests

The authors declare no competing interests.
