## [Peer Review File · Communications Biology]

Reviewers' comments:

Reviewer #1 (Remarks to the Author):

Shi et al. present the structure of the T4 Spackle protein bound to the gp5 lysozyme domain. They also show that the Spackle protein inhibits the gp5 lysozyme, but not the T4 endolysin. The experiments appear to be well-done and the conclusions are interesting, but I did not fully review the manuscript due to the following two reasons:

1. Although I do commend the authors for including their related manuscript describing the Spackle-alone structure submitted to Acta Cryst D, I don't understand the reasoning of submitting the Spackle-alone structure somewhere else (at about the same time) and the complex structure here. Together, they would make for a stronger manuscript and for more convenience for the reader.

2. Coordinate and structure factor files should be provided for proper judgement of this paper (not just PDB validation files). I don't know if this is journal policy, but in my opinion it should be.

Minor comments:

Are HEPES molecules in endolysins/lysozyme crystal structures a more common feature? There seem to be more of them in the PDB. Could this be interesting to comment on? Does HEPES somehow mimic the substrate?

Figure 6: Spakle => Spackle

Reviewer #2 (Remarks to the Author):

In this paper, Shi et al report the structure and function of the bacteriophage T4 Spackle protein, gp61.3. Previous genetic studies suggested that the spackle protein binds to the lysozyme domain of the spike protein, gp5, and inhibits its lysozyme activity. Expressed as an early gene product, the spackle protein is thought to locate in the periplasm and by binding to gp5 spike protein of another phage trying to enter the same E. coli, prevents the second infection, a phenomenon known as superinfection.

The authors confirmed the above hypothesis by biochemical experiments, first by showing that the purified spackle protein and gp5 lysozyme domain form a 1:1 complex, and then by in vitro enzymatic assays that showed complete inhibition of gp5 lysozyme at 2:1 ratio of spackle protein to gp5 lysozyme.

Furthermore, most importantly, the authors determined the structure of the spackle:gp5 lysozyme domain complex by X-ray crystallography. The structure showed that the spackle protein binds near the oligosaccharide-binding cleft of the lysozyme, distant from the catalytic center. This binding likely interferes with substrate binding and further, it appears that there is a conformational change that leads to changes in the orientation of the catalytic residues. These provide a structural and mechanistic basis for spackle protein function.

This is a nice piece of work and clarifies the mechanism of a significant phenomenon that remained unresolved for a long time. The paper is well written and the data are explained well.

One suggestion I have for the authors is to discuss any evolutionary information on the spackle protein: i) is it present in other T4 family bacteriophages? Does it show up in other phages in the phage databases? How well is it conserved and any interesting mutations that shed further light on function? are there related sequences or folds present in the biological kingdom?

Reviewer #3 (Remarks to the Author):

This is an interesting paper that settles a long-standing issue of T4 biology at the structural level. I am not a crystallographer so I did not read the technical aspects of the structure determination critically. Most of my comments have to do with language, style and emphasis. My major reservation has to do with the interpretation of Fig. 5B (see comments on line 114 below) and, most importantly, the question posed at the very end of the MS, which I do not find credible. (See text wrt line 162 below).

34 The treatment of gp5 processing is confusing. Please state clearly how gp5 is processed. I thought there were two cleavages, one occurring at assembly and another during penetration.

13 and 39ff "resistance" traditionally means something very specific in phage biology, usually the absence of irreversible adsorption. Perhaps "insensitivity"?

23ff I am not sure that the typical reader will understand which lysozyme is meant here? It is not clear that the endolysin is biologically inhibited. Gp5 lysozyme is not a classic enzyme in biology.

35 "the" cell wall

43 proteolytic cleavage

46 is glycanhydrolase the official Enzyme Commission term?

47ff: this segment is all Results and should not be in the Introduction.

71ff and elsewhere.

The authors use gp5 lysozyme rather colloquially. I think the 20 kDa fragment is the "gp5 lysozyme domain". In contrast, gpe is the endolysin, which, like the 20 kDa domain, has "true lysozyme" activity. Also, I urge that this terminology, which is less confusing, be adopted throughout this paper.

88ff: what makes a contact "key"? Doesn't this have to be determined by binding equilibria experiments with wt and mutants?

102-103: "appears to stabilize". Structures do not speak to thermodynamics. Possibly soften this phrase.

106ff The partial disulfide bond argument seems a bit unusual. Is the binding done under conditions that favor reduction? Is it possible that an equilibrium is going on in the solution state and the reduced form gets trapped in the complex preferentially? Can the authors cite other examples in the crystallographic literature?

114ff Lysozyme assays are difficult to interpret because of the lack of a defined substrate. While I don't disagree with the conclusions, the authors should temper their conclusions by acknowledging the problems with inability to determine enzymatic parameters. When I look at Fig. 5B, I don't see any difference in the slopes of the quenching assays. Instead, there appears to be a delay in the degradation dependent on the ratio of spackle to the endolysin. Impossible to interpret and the authors should say so. I see no evidence of spackle inhibition of the endolysin.

117ff: This segment points out the importance of qualifying interpretation of the assays, since the only thing that can be observed with an undefined substrate is release of degradation product. We have no idea how the three muralytic enzymes bind to the substrate used. It might be different between them.

121 "Consistently" is inappropriate. As written, it means "reproducibly".

127 Discussion (singular)

135 Why "critical"?

157 The amber revertant does not "cause" the amino acid change. Reword.

160ff: Suddenly the use of T4L! Substitute T4 gpe or endolysin.

162: This makes no sense. As I understand it, the timing of the bursting depends on the gpt holin. According to present models, the endolysin is never in the periplasm until the holin disrupts the membrane and allows it to attack the PG. If Spackle were to inhibit the endolysin, it would be suicidal for T4, because no new phage virions would accumulate and all that would be accomplished is a slower release of the same number of phage progeny. Are there no single step growth experiments with spackle mutants? This would answer this question, since under the conditions of single step growth, lysis inhibition would not apply, so the timing of bursting would be detectably accelerated if spackle was not present.

163 phage-infected

We thank all three expert reviewers for their time and constructive suggestions. Our responses to their comments are in blue.

Referee expertise:

Referee #1: X-ray crystallographer/Viral proteins

Referee #2: T4 bacteriophage biology

Referee #3: Phage biology

Reviewers' comments:

Reviewer #1 (Remarks to the Author):

Shi et al. present the structure of the T4 Spackle protein bound to the gp5 lysozyme domain. They also show that the Spackle protein inhibits the gp5 lysozyme, but not the T4 endolysin. The experiments appear to be well-done and the conclusions are interesting, but I did not fully review the manuscript due to the following two reasons:

1. Although I do commend the authors for including their related manuscript describing the Spackle-alone structure submitted to *Acta Cryst D*, I don't understand the reasoning of submitting the Spackle-alone structure somewhere else (at about the same time) and the complex structure here. Together, they would make for a stronger manuscript and for more convenience for the reader.

The structure of free Spackle has been published in *Acta Cryst D* (PMID: 32876065), where the paper was more focused on technical aspects. We agree that the two papers could have been consolidated.

2. Coordinate and structure factor files should be provided for proper judgement of this paper (not just PDB validation files). I don't know if this is journal policy, but in my opinion it should be.

We have provided the pdb and mtz files as per this request.

Minor comments:

Are HEPES molecules in endolysins/lysozyme crystal structures a more common feature? There seem to be more of them in the PDB. Could this be interesting to comment on? Does HEPES somehow mimic the substrate?

There are many lysozyme structures in the PDB and some of them indeed have bound HEPES. We found that the HEPES molecule in the oligosaccharide-binding cleft in our structure roughly matches that bound in one of the wild type T4 lysozyme entries (PDB ID: 4S0W). These HEPES molecules partially overlap with a pyranose ring in the substrate-mimic-bound T4 lysozyme structure (PDB ID: 148L). That said, HEPES does not appear to mimic the substrate, beyond binding in the same general location.

Figure 6: Spakle => Spackle

The typo has been fixed.

Comments on the coordinate and structure factor files:

I would like to thank the authors for providing the files. The P21 structure (code 6XC0) looks to have been very well refined. The P212121 structure (code 6XC1) could be improved a bit. There are some densities where additional solvent molecules could be placed, such as near the backbone N atom of Trp-A-316, but also some other places. For some amino acids, a different rotamer might have been a better choice, such as Glu-C-30 and Arg-C-56, which are quite clear, but also perhaps Lys-C-87 and Lys-C-53 (I didn't check every amino acid). At the same time, I have to admit none of these small improvements would affect the biological conclusions of the paper.

We thank the reviewer for bringing up these points to improve the model. We have accordingly updated and re-refined the structure in the orthorhombic crystal system, which lowered the free R-factor from 21.7 to 21.5 %. Table 1 stats have been updated to reflect these changes in the coordinates.

Reviewer #2 (Remarks to the Author):

In this paper, Shi et al report the structure and function of the bacteriophage T4 Spackle protein, gp61.3. Previous genetic studies suggested that the spackle protein binds to the lysozyme domain of the spike protein, gp5, and inhibits its lysozyme activity. Expressed as an early gene product, the spackle protein is thought to locate in the periplasm and by binding to gp5 spike protein of another phage trying to enter the same *E. coli*, prevents the second infection, a phenomenon known as superinfection.

The authors confirmed the above hypothesis by biochemical experiments, first by showing that the purified spackle protein and gp5 lysozyme domain form a 1:1 complex, and then by in vitro enzymatic assays that showed complete inhibition of gp5 lysozyme at 2:1 ratio of spackle protein to gp5 lysozyme.

Furthermore, most importantly, the authors determined the structure of the spackle:gp5 lysozyme domain complex by X-ray crystallography. The structure showed that the spackle protein binds near the oligosaccharide-binding cleft of the lysozyme, distant from the catalytic center. This binding likely interferes with substrate binding and further, it appears that there is a conformational change that leads to changes in the orientation of the catalytic residues. These provide a structural and mechanistic basis for spackle protein function.

This is a nice piece of work and clarifies the mechanism of a significant phenomenon that remained unresolved for a long time. The paper is well written and the data are explained well.

One suggestion I have for the authors is to discuss any evolutionary information on the spackle protein: i) is it present in other T4 family bacteriophages? Does it show up in other phages in the phage databases? How well is it conserved and any interesting mutations that shed further light on function? are there related sequences or folds present in the biological kingdom?

Spackle is found in the T-even series (T2/T4/T6) as well as other bacteriophages. We compared the sequences of a total of 178 Spackle homologs available in the UniProt database and summarized the result in the new **Supplementary Fig. S6**. Although the utility of this information is debatable due to a limited sequence diversity, we confirmed that most of the residues positioned close to gp5 lysozyme in the complex are highly conserved, suggesting a similar mode of lysozyme interaction for these Spackle homologs.

We found through a DALI search that the helical bundle fold of Spackle has modest structural similarity to (small portions of) functionally unrelated eukaryotic proteins. These similarities appear to be superficial and most likely do not have any functional relevance (please see Supplementary Fig. S3 of our *Acta Cryst D* paper, PMID: 32876065).

Reviewer #3 (Remarks to the Author):

This is an interesting paper that settles a long-standing issue of T4 biology at the structural level. I am not a

crystallographer so I did not read the technical aspects of the structure determination critically. Most of my comments have to do with language, style and emphasis. My major reservation has to do with the interpretation of Fig. 5B (see comments on line 114 below) and, most importantly, the question posed at the very end of the MS, which I do not find credible. (See text wrt line 162 below).

34 The treatment of gp5 processing is confusing. Please state clearly how gp5 is processed. I thought there were two cleavages, one occurring at assembly and another during penetration.

We apologize for the confusion. Gp5 is processed during assembly of the baseplate but the C-terminal ~20 kDa fragment remains associated with the tail of the phage. Upon infection, the C-terminal fragment is released from the baseplate and the gp5 lysozyme domain is activated. We have accordingly corrected the sentence referring to the proteolytic cleavage of gp5.

13 and 39ff “resistance” traditionally means something very specific in phage biology, usually the absence of irreversible adsorption. Perhaps “insensitivity”?

The term “resistance” is used broadly in discussing various antiviral mechanisms of bacteria including superinfection exclusion (e.g. <https://www.nature.com/articles/nrmicro2315>). Also, the phenotype of spackle mutant was originally described as an “inability to develop resistance to lysis from without” (e.g. <https://pubmed.ncbi.nlm.nih.gov/4589853/>). Thus, we feel that it might be okay to use the term resistance in these contexts.

23ff I am not sure that the typical reader will understand which lysozyme is meant here? It is not clear that the endolysin is biologically inhibited. Gp5 lysozyme is not a classic enzyme in biology.

We updated the misleading sentence to the following to make it clearer that we are referring to lysozyme in general.

“Our work reveals a novel mode of inhibition for lysozyme, a widespread class of enzymes in biology, and provides mechanistic understanding....”

35 “the” cell wall

The sentence was fixed. We thank the reviewer for catching this and other grammatical problems.

43 proteolytic cleavage

The typo was fixed.

46 is glycanhydrolase the official Enzyme Commission term?

The term “glycanhydrolase” has been changed to ‘glycoside hydrolase’, which would be more appropriate.

47ff: this segment is all Results and should not be in the Introduction.

We agree, but the formatting guide of *Communications Biology* says that the final paragraph of Introduction should be a brief summary of the major results and conclusions.

71ff and elsewhere.

The authors use gp5 lysozyme rather colloquially. I think the 20 kDa fragment is the “gp5 lysozyme domain”. In contrast, gpe is the endolysin, which, like the 20 kDa domain, has “true lysozyme” activity. Also, I urge that this terminology, which is less confusing, be adopted throughout this paper.

We agree and think that the 20 kDa fragment would most accurately be described as ‘the lysozyme domain of gp5’. Thus, “gp5 lysozyme” was changed to “the lysozyme domain of gp5” in Abstract and at one location in Introduction. We also made the change in the first paragraph of Result section and define the term ‘gp5 lysozyme’ as following.

When the purified Spackle (9.7 kDa) and the lysozyme domain of gp5 (20.1 kDa, hereinafter referred to as gp5 lysozyme) were individually injected...

88ff: what makes a contact “key”? Doesn’t this have to be determined by binding equilibria experiments with wt and mutants?

We changed “key” to “direct”, which is more objective.

102-103: “appears to stabilize”. Structures do not speak to thermodynamics. Possibly soften this phrase.

We changed “appears to stabilize” to “could stabilize”.

106ff The partial disulfide bond argument seems a bit unusual. Is the binding done under conditions that favor reduction? Is it possible that an equilibrium is going on in the solution state and the reduced form gets trapped in the complex preferentially? Can the authors cite other examples in the crystallographic literature?

The crystals used in this study were grown in a non-reducing condition but it’s possible that a some fraction of the protein in the reduced form was preferentially trapped in the complex during crystallization. We have revised the sentence to mention this possibility.

114ff Lysozyme assays are difficult to interpret because of the lack of a defined substrate. While I don’t disagree with the conclusions, the authors should temper their conclusions by acknowledging the problems with inability to determine enzymatic parameters. When I look at Fig. 5B, I dont see any difference in the slopes of the quenching assays. Instead, there appears to be a delay in the degradation dependent on the ratio of spackle to the endolysin. Impossible to interpret and the authors should say so. I see no evidence of spackle inhibition of the endolysin.

We have revised the description of lysozyme activity inhibition data as following, tempering our conclusions by acknowledging the problems with inability to determine enzymatic parameters.

On the other hand, in parallel experiments Spackle also affected the activity of cytoplasmic T4 lysozyme (hereinafter referred to as endolysin), which shares 43% sequence identity with the lysozyme domain of gp5²³ (Figs. 5b, 6). However, the addition of Spackle did not significantly alter the overall rate of product accumulation by endolysin and instead led to a longer initial lag phase, the reason of which is unknown. In any case, the inhibition of endolysin by Spackle, if any, was weak, where we still observed significant enzyme activity even at 4-fold excess of Spackle over endolysin (Fig. 5b).

117ff: This segment points out the importance of qualifying interpretation of the assays, since the only thing that can be observed with an undefined substrate is release of degradation product. We have no idea how the three muralytic enzymes bind to the substrate used. It might be different between them.

We agree that the three muralytic enzymes might have different modes of interaction with the substrate, which may account for their varying basal (uninhibited) activities.

121 “Consistently” is inappropriate. As written, it means “reproducibly”.

We think that the weak (or no) inhibition of T4 endolysin activity by Spackle is consistent with the observation that these proteins do not form a stable complex as was observed for gp5 lysozyme domain and Spackle.

127 Discussion (singular)

The header was corrected to Discussion.

135 Why “critical”?

The sentence has been revised to:

These residues correspond to $\alpha 5$ helix and the following C-terminal tail, which are shown here to make extensive interaction with gp5 lysozyme (Figs. 2, 3).

157 The amber revertant does not “cause” the amino acid change. Rerword.

The confusing sentence was revised to:

Notably, a temperature sensitive pseudo-revertant of an amber mutation in gene e was isolated and the reversion mutation was mapped in gene 5 (5ts1), which leads to the Gly322 to Asp amino acid substitution in gp5 lysozyme.

160ff: Suddenly the use of T4L! Substitute T4 gpe or endolysin.

We changed T4L to endolysin throughout the manuscript including figures.

162: This makes no sense. As I understand it, the timing of the bursting depends on the gpt holin. According to present models, the endolysin is never in the periplasm until the holin disrupts the membrane and allows it to attack the PG. If Spackle were to inhibit the endolysin, it would be suicidal for T4, because no new phage virions would accumulate and all that would be accomplished is a slower release of the same number of phage progeny.

Discussion has been extensively revised to discount the possibility that Spackle has a role to modulate the activity of T4 endolysin.

Are there no single step growth experiments with spackle mutants? This would answer this question, since under the conditions of single step growth, lysis inhibition would not apply, so the timing of bursting would be detectably accelerated if spackle was not present.

We are not aware of the availability of such data.

163 phage-infected

The sentence has been removed.